# The Classification of VOCs Based on Sensor Images Using a Lightweight Neural Network for Lung Cancer Diagnosis

**DOI:** 10.3390/s24092818

**Published:** 2024-04-28

**Authors:** Chengyuan Zha, Lei Li, Fangting Zhu, Yanzhe Zhao

**Affiliations:** Department of Electronics and Electrical Engineering, Changchun University of Technology, Changchun 130012, China; cha0521@foxmail.com (C.Z.); zft4867@163.com (F.Z.); yanzhezhao19980502@163.com (Y.Z.)

**Keywords:** breath detection, sensor arrays, convolutional neural network, lightweight neural network, cancer detection

## Abstract

The application of artificial intelligence to point-of-care testing (POCT) disease detection has become a hot research field, in which breath detection, which detects the patient’s exhaled VOCs, combined with sensor arrays of convolutional neural network (CNN) algorithms as a new lung cancer detection is attracting more researchers’ attention. However, the low accuracy, high-complexity computation and large number of parameters make the CNN algorithms difficult to transplant to the embedded system of POCT devices. A lightweight neural network (LTNet) in this work is proposed to deal with this problem, and meanwhile, achieve high-precision classification of acetone and ethanol gases, which are respiratory markers for lung cancer patients. Compared to currently popular lightweight CNN models, such as EfficientNet, LTNet has fewer parameters (32 K) and its training weight size is only 0.155 MB. LTNet achieved an overall classification accuracy of 99.06% and 99.14% in the own mixed gas dataset and the University of California (UCI) dataset, which are both higher than the scores of the six existing models, and it also offers the shortest training (844.38 s and 584.67 s) and inference times (23 s and 14 s) in the same validation sets. Compared to the existing CNN models, LTNet is more suitable for resource-limited POCT devices.

## 1. Introduction

Due to its high incidence and lethality, lung cancer (LC) imposes a significant burden on the healthcare system [1,2]. By the end of 2023, it is forecasted that there will be 609,820 cancer-related deaths in the United States, with lung cancer remaining the leading cause of cancer mortality among them [3]. Currently, clinical detection of lung cancer primarily relies on cellular or histopathology examinations, radiographic imaging such as X-rays and CT scans, and tumor marker assays in bodily fluids. But, these existing detection techniques have obvious drawbacks, including high cost and significant harm to the human body.

Volatile organic compounds (VOCs) contained in exhaled human breath are closely associated with various diseases. Breath analysis was one solution to be put forward, which is a non-invasive early screening method that can be employed for the screening of various diseases, including lung cancer, diabetes, breast cancer and so on [4,5,6] Research has indicated that lung cancer results in elevated levels of acetone and ethanol in exhaled breath. Both of these gases can serve as reliable exhalation biomarkers for early lung cancer [7,8]. In current clinical trials, spectroscopic methods, mass spectrometry, chromatography-related techniques, and electronic noses (e-nose) are considered as relatively viable and efficient standard technologies for detecting VOCs in human exhaled breath [9,10,11]. However, techniques such as mass spectrometry, chromatography and spectroscopic methods are constrained by their high equipment costs and technical operator requirements. In contrast, gas sensor array-based e-nose gas sensing technology holds greater cost advantage and development potential for application in breath analysis [12,13,14].

Through the analysis of the data collected by the sensor array by the pattern recognition algorithm, the electronic nose can effectively identify complex gases. In a pattern recognition task, feature selection and feature extraction directly affect the detection performance of electronic nose. Marzorati et al. [15] extracted nine features from the response of each gas sensor to exhaled gas. Liu et al. [16] used 19 sensors, selected 13 composite features from each sensor, and combined with classical classifiers to verify the feasibility of identifying LC patients through VOCs. In previous studies, it has been found that the whole feature extraction process is particularly complex, and it is necessary to try the feature extraction method continuously to obtain better results. Gramian Angular Field (GAF) is a data visualization method, proposed by Wang et al., which converts time series into two-dimensional color images [17]. GAF encodes one-dimensional time series into two-dimensional color images with more prominent key features, so as to display the important information hidden in the sensor signal more clearly. By using the data visualization technology of GAF, the original data do not need to be processed, and are directly converted into two-dimensional color images, which can not only retain the deep features of the signal, but also avoid complex feature extraction engineering [18].

Currently, pattern recognition algorithms employed in electronic noses are categorized into classical gas identification algorithms (machine learning), artificial neural networks (ANNs), and biologically inspired pulse neural networks (SNNs) [19,20,21]. To address complex gas recognition tasks, ANNs have been regarded as the current popular choice [22,23]. Compared with machine learning and SNNs, ANNs exhibit strong adaptability and do not require model redesign for different training tasks. Additionally, the nets can automatically learn complex features from data without the need for manual feature extraction and they encompass various neural network architectures, such as backpropagation neural networks (BPNNs), convolutional neural networks (CNNs), etc. Avian et al. [24]. proposed a CNN architecture and built two models for analyzing VOCs in exhaled gas. The first model receives the signal processed by different feature extraction methods as input, while the second model directly processes the original signal. The results indicate that different classifiers demonstrate varying effects depending on the employed feature extraction methods, with kernel PCA (KPCA) showing a positive impact on performance. Guo et al. [25] introduced an innovative deep learning framework that combines an electronic nose to predict odor descriptor ratings, which was the first application of convolutional long short-term memory (ConvLSTM) on an electronic nose for olfaction prediction.

Although convolutional neural networks have found extensive application in gas classification, they possess a substantial number of trainable parameters, high computational complexity, slow inference speed and are not hardware-friendly for devices, which hinder the transportability of pattern recognition algorithms into embedded systems. To address this issue, convolutional neural networks have increasingly evolved towards lightweight architectures [26,27].

The contributions of this article are summarized as follows:(1)A hardware-friendly lightweight neural network model (LTNet) using a depth-separable convolution structure for gas classification is constructed.(2)To settle the decrease in classification accuracy caused by depthwise separable convolutions, we propose to add squeeze-and-excitation (SE) attention mechanisms and residual connections in the model.(3)The convolutional and batch normalization (BN) layers are combined together so as to reduce the model parameters, speed up the inference speed and improve the stability of the model.(4)Compared to the unimproved LTNet (LTNet (Original version)), this validates the effectiveness of the improvements made to LTNet.

## 2. Experimental Section

Data collection was performed using the CGS-8 intelligent gas sensor analysis system provided by Beijing Ailite Technology Co., Ltd., a company located in Beijing, China. Subsequently, the collected data were transformed into images using the Gramian Angular Field (GAF), which served as valid inputs to LTNet. Two different datasets were employed to assess the performance of the model, and the overall process is illustrated in Figure 1.

### 2.1. Data Source I: Gas Mixture Dataset

Exhaled breath from the lung cancer patients contains biomarkers, including numerous species, such as acetone, ethanol, isoprene and etc. By detecting the types and concentrations of these biomarkers, we can assess changes in physiological state in vivo and achieve early screening for lung cancer. To validate LTNet’s classification capabilities, the study utilized acetone and ethanol gases to simulate breath biomarkers found in lung cancer patients. In the experiment, a sensor array composed of 16 commercial semiconductor metal oxide gas sensors manufactured by FIGARO was used, which matched the sensor model in Data Source II (UCI database).

The experiment employed a static gas volumetric method using 98% AR acetone and 98% anhydrous ethanol with a microsyringe capable of handling a range of 10 μL for liquid extraction. As indicated by Equation (1), it can be observed that conducting experiments directly with high-concentration test liquids would result in a very small volume of extracted liquid, making it difficult to inject into the gas chamber. To address this issue, the concentration of the test liquid was chosen to dilute to 10% in the experiment.
(1)$$Q=\frac{V\times C\times M}{22.4\times d\times r}\times 10^{-9}\times \frac{273+T_{R}}{273+T_{B}}$$
where Q is the volume of the test liquid (mL), V denotes the volume of the gas chamber (mL), C stands for the desired gas concentration to be prepared (ppm), M is the molecular weight of the substance, d is the concentration of the test liquid, r signifies the liquid density ($g/{\mathrm{cm}}^{3}$), $T_{R}$ represents the laboratory ambient temperature (°C) and $T_{B}$ is the gas chamber temperature (°C).

Following the specifications in the sensor manual, the working voltage of all 16 sensors was set to 5 V. The sensor operating current was adjusted on the CGS-8 smart gas sensing analysis system through multiple experiments to identify the optimal operating current for each sensor. The sensor models and their optimal operating currents are presented in Table 1, respectively.

After setting the optimal operating current, it is necessary to preheat the sensor for two hours and wait for the baseline to stabilize. Then, the evaporation and heating functions of the experimental apparatus are activated. A microsyringe is used to extract a certain amount of liquid from the test liquid prepared according to Formula (1), which is then vertically dropped into the evaporation dish. The sensor array is exposed to acetone, ethanol or a binary mixture of these two VOC gases. The concentration indices for the two gas mixtures are detailed in Table 2. The experimental response time is set approximately 120 s, and the recovery time is also about 120 s.

### 2.2. Data Sources II: UCI Database

This work also used a public database from the University of California (UCI) to complement and validate Data Source I. This dataset is a collection of gas sensor drift datasets at different concentrations, collected by the Chemical Signaling Research Laboratory at the UCI BioCircuits Institute in San Diego [28,29]. The acetone concentrations in the dataset range from 12–500 ppm and ethanol concentrations range from 10–500 ppm. A total of 4650 datapoints from UCI dataset were used for classification with LTNet.

### 2.3. Experimental Environment and Hardware Configuration

The algorithmic programming environment for this study is Python 3.10, running on a computer with an RTX 3060 graphics card. The LTNet network, as well as the comparison network, uses the Adam optimizer with a cross-entropy loss function. The model usually converges after about ten rounds of training, so epoch was set to 30. Conventional convolutional neural networks take up a lot of graphics card memory during training, and for comparison purposes, the batch size was set to 16. For the own mixed gas dataset, the learning rate was set to 0.0006, while for the UCI database, the learning rate was set to 0.0004.

## 3. Data Processing

### 3.1. Image Conversion Methods

CNNs are typically used for processing two-dimensional image data. However, the raw data collected from each channel of the sensor array were one-dimensional results and not suitable for processing by the CNNs directly. To figure out this issue, an image transformation model based on Gramian Angular Field (GAF) was utilized [30], transforming one-dimensional time series data into two-dimensional images that would become effective inputs for the LTNet. The coding diagram based on GAF is shown in Figure 2. The response data of 16 sensors were selected at a certain sampling point, and the response data were denoted as $X=\left\{x_{1},\ldots ,x_{16}\right\}$,then normalization of the response data X to the range of $\left[-1,1\right]$ was carried out using min–max normalization with Formula (2).
(2)$${\tilde{x}}_{i}=\frac{\left(x_{i}-\max(X))+(x_{i}-\min(X)\right)}{\max(X)-\min(X)},i=1,2\ldots 16$$
where X represents the response data of the sampling points and $x_{i}$ is the response data of the ith sensor. This step constrains the angular range between 0 and π, facilitating the acquisition of more detailed GAF information.

The selected data for this study consist of the response data from 16 sensors at a specific sampling point, and do not involve time series. Therefore, there is no need to encode the timestamps as radii. Formula (3) was used to calculate the arccosine values of the response data for the sensors at this sampling point.
(3)$$\phi =\arccos({\tilde{x}}_{i}),-1\leq {\tilde{x}}_{i}\leq 1,{\tilde{x}}_{i}\in X$$

After transforming the scaled ${\tilde{x}}_{i}$ into polar coordinates, the correlation between response data of different sensors was captured through the summation of triangular relationships among each point. Therefore, GASF and GADF were defined by the following equations, respectively.
(4)$$GADF=\left(\begin{array}{ccc} \sin(\phi_{1}-\phi_{1}) & \ldots & \sin(\phi_{1}-\phi_{n})\\ \vdots & \vdots & \vdots \\ \sin(\phi_{n}-\phi_{1}) & \cdots & \sin(\phi_{n}-\phi_{n}) \end{array}\right)$$
(5)$$GASF=\left(\begin{array}{ccc} \cos(\phi_{1}+\phi_{1}) & \ldots & \cos(\phi_{1}+\phi_{n})\\ \vdots & \vdots & \vdots \\ \cos(\phi_{n}+\phi_{1}) & \cdots & \cos(\phi_{n}+\phi_{n}) \end{array}\right)$$

In which, $\phi_{1}$ and $\phi_{n}$ represent the normalized and inverse cosine-transformed response data of the first and nth sensors, respectively, in the sensor array.

### 3.2. Lightweight Neural Network Model

The high complexity, large number of parameters and relatively slow inference speed of neural networks could potentially impede the feasibility of porting pattern recognition algorithms to embedded systems. Therefore, a lightweight neural network model (LTNet) is proposed to figure out this problem. It includes a backbone network based on depthwise separable convolutions, and the complete network architecture is illustrated in Figure 3.

As shown in Figure 3, the architecture of the LTNet is mainly composed of the ConvBN layer and the LTBlock module. The convolutional layer is fused with the BN layer into a new ConvBN layer and the weights and biases of the ConvBN layer are reinitialized. The aim of this design is to reduce the parameter size of the network and improve the inference speed of the network in the validation procedure.

To extract features from the input image, the network uses a 3 × 3 deep convolution to learn the feature maps of the input channels when the input image passes through the ConvBN layer, in order to preserve the correlation between different channels. Subsequently, the feature maps produced by deep convolution are mapped by a 1 × 1 point-by-point convolutional layer to improve the ability to capture local information of the models. In order to avoid increasing the depth of the network, the LTBlock module uses the ConvBN layer for feature extraction several times, and introduces the SE attention mechanism and residual connectivity in order to enhance the feature interactions between channels, maintaining the integrity of the original input information. The LTBlock module employs the hard swish as activation function (Hswish) in order to introduce the nonlinear nature of the output of the network neurons, and finally obtains the classification results through the fully connected layer (FC layer).

### 3.3. Calculation of Depthwise Separable Convolutions Parameters

To enhance the efficiency of standard convolutions while maintaining network performance and generalization capability, LTNet introduces depthwise separable convolutions. Depthwise separable convolutions decompose standard convolutions into two steps: firstly, a depthwise convolution with a K × K kernel, followed by a pointwise convolution with a 1 × 1 kernel. In the depthwise convolution stage, independent convolution filters are applied to each input channel, making the convolution operation independent in the channel dimension and effectively capturing spatial features within each channel [31]. The role of pointwise convolution is to construct new features by computing a linear combination of input channels. The parameters and floating-point operations (FLOPs) for standard convolution and depthwise separable convolution are as follows:(6)$$C_{P}=K\times K\times C_{\mathrm{in}}\times C_{\mathrm{out}}$$
(7)$$C_{F}=H\times W\times K\times K\times C_{in}\times C_{out}$$
(8)$$D_{P}=K\times K\times C_{in}+C_{in}\times C_{out}$$
(9)$$D_{F}=H\times W\left(K\times K\times C_{in}+C_{in}\times C_{out}\right)$$
where $C_{P}$ and $C_{F}$ represent the parameters and floating-point operations (FLOPs) of standard convolution, and $D_{P}$ and $D_{F}$ represent the parameters and FLOPs of depthwise separable convolution. The convolution kernel size is K×K. H and W represent the dimensions of the output feature map.$C_{in}$ is the number of input feature map channels, and $C_{out}$ is the number of output feature map channels. The comparison of parameters and FLOPs between standard convolution and depthwise separable convolution is as follows:(10)$$\frac{D_{P}}{C_{p}}=\frac{K\times K\times C_{in}+C_{in}\times C_{out}}{K\times K\times C_{\mathrm{in}}\times C_{\mathrm{out}}}=\frac{1}{C_{\mathrm{out}}}+\frac{1}{K\times K}$$
(11)$$\frac{D_{F}}{C_{F}}=\frac{H\times W\left(K\times K\times C_{in}+C_{in}\times C_{out}\right)}{H\times W\times K\times K\times C_{in}\times C_{out}}=\frac{1}{C_{out}}+\frac{1}{K\times K}$$

From Equations (10) and (11), it can be observed that depthwise separable convolution involves fewer parameters and floating-point operations, making the model more lightweight.

This study integrates deep convolution with the BN layer, further reducing the number of parameters in the LTNet model and accelerating the model’s inference speed. The parameter calculations before and after fusion are shown in Equations (12) and (13).
(12)$$U_{p}=C_{out}\times C_{in}\times K\times K+4\times C_{out}$$
(13)$$F_{p}{=C}_{out}\times C_{in}\times K\times K+C_{out}$$
where $U_{p}$ is the number of parameters before fusion and $F_{p}$ is the number of parameters after fusion. The parameters of the BN layer are mainly determined by four parameters, which are scale parameter, offset parameter, mean and variance of BN layer. The purpose of the BN layer is to normalize the data on each channel, so the number of parameters of the BN layer corresponds to the output channel $C_{out}$. From Equations (12) and (13), fusing the depth-separable convolution with the BN layer only reduces the parameters of $C_{out}$ with a factor of three. But, the fused weights and biases can be directly used in the testing and validation phases of LTNet, which can speed up the inference of LTNet and reduce the computation and memory consumption. The merged ConvBN layer obtained after fusion needs to be computed according to the relevant parameters of the convolutional and BN layers to generate new weights and biases, which are described by the following formula:(14)$$\begin{array}{cc} Y_{f} & =\frac{\gamma \cdot \left(X-\mu \right)}{\sqrt{\sigma^{2}+\varepsilon }}+\beta \\ & =\frac{\gamma \cdot \left(X_{c}W_{c}+b_{c}\right)-\mu }{\sqrt{\sigma^{2}+\varepsilon }}+\beta \\ & =\frac{\gamma \cdot W_{c}\cdot X_{c}}{\sqrt{\sigma^{2}+\varepsilon }}+\frac{\gamma \cdot (b_{c}-\mu )}{\sqrt{\sigma^{2}+\varepsilon }}+\beta \end{array}$$
where $Y_{f}$ represents the output of the merged ConvBN layer, X is the output of the pre-merged convolutional layer, γ and β denote the weights and biases of the BN layer, $\sigma^{2}$ represents the variance and ε is a constant that prevents division by zero. $X_{c}$ represents the input of the pre-merged convolutional layer, while $W_{c}$ and $b_{c}$ are the weights and biases of the convolutional layer, respectively. Formula (14) can be simplified as follows:(15)$$Y_{f}=W_{f}\cdot X_{c}+b_{f}$$
where the weights $W_{f}$ and biases $b_{f}$ of the ConvBN layer are calculated as follows:(16)$$W_{f}=\frac{\gamma \cdot W_{c}}{\sqrt{\sigma^{2}+\varepsilon }}$$
(17)$$b_{f}=\frac{\gamma \cdot (b_{c}-\mu )}{\sqrt{\sigma^{2}+\varepsilon }}+\beta$$

## 4. Results and Discussion

The dataset was divided into training, testing and validation sets in a 6:3:1 ratio. For the own mixed gas dataset, the numbers of images in the training, testing and validation sets were 4474, 2234 and 744, respectively. For the UCI database, the training, testing and validation sets consisted of 2791, 1395 and 464 images, respectively. In this study, validation set accuracy and six evaluation metrics were adopted as criteria for assessing both the lightweight of the model and its classification performance. These criteria include the model’s total accuracy on the validation set (accuracy) (ethanol, acetone and the mixture), the time required for the model to complete thirty training epochs (training time), the GPU memory usage under the same conditions when different models are trained with a cleared background (GPU RAM), the inference time on the validation set (Inference time), the model’s parameters (params) and the size of the best-preserved weights on the test set (weight size).

### 4.1. Data Conversion Comparison Test and Results Discussion

In the data preprocessing, GASF is compared with Gramian Angular Difference Field (GADF), Short-Time Fourier Transform (STFT) and Markov Transition Field (MTF). Figure 4 illustrates the images transformed by these four data transformation methods.

LTNet was employed to evaluate four different methods for transforming one-dimensional time series into two-dimensional images. The confusion matrices, which can display the classification of each sample intuitively and make up an important index used to measure the classification performance, are depicted in Figure 5a–d, while Table 3 presents a comparative analysis of the results obtained using these four methods.

From Table 3 and the results of confusion matrices, it is evident that STFT exhibits the highest total classification accuracy, reaching 100%. The majority of errors for GADF and GASF are concentrated in the mixed gas class, with total classification accuracies of 98.79% and 99.06%, respectively. MTF, on the other hand, experiences more classification errors in the mixed gas and ethanol classes, resulting in the total accuracy of only 92.34%. However, training with images transformed using STFT takes the longest training time, reaching 1630.92 s, and it also requires the highest GPU RAM usage. Conversely, training with images transformed using GASF for the same thirty epochs only takes 844.38 s and occupies a mere 2.1 GB of GPU RAM, while achieving similar accuracy to STFT. Therefore, GASF was chosen as the data transformation method in this work.

### 4.2. Model Evaluation and Comparison Experiment

LTNet is compared with a total of six different networks, including three traditional convolutional neural networks (AlexNet, ResNet50, VGG16) and two lightweight convolutional neural networks (EfficientNet and MobileNetV3_large), as well as the unimproved version of LTNet (LTNet (Original version)). AlexNet is a relatively deep neural network, which facilitates the model in learning more complex features. ResNet50 implements classification using skip-connected residual blocks, VGG16 employs deep convolutional networks for feature extraction from raw samples, followed by classification using fully connected layers. EfficientNet achieves high performance in resource-constrained environments through strategies like compound scaling and width multiplier. MobileNetV3_large classifies raw samples using depthwise separable convolution. To verify the effectiveness of the improvements on LTNet, LTNet (Original version) was used to compare with LTNet. LTNet (Original version) refers to a version of LTNet that does not utilize deep separable convolution, the SE attention mechanism, residual connections or the fusion of convolution layers and BN layers. Instead, it solely implements the network architecture of LTNet.

Before conducting the classification task, we compared the model parameters and the size of the best saved weights during training of LTNet with the other six networks, as shown in Table 4. Among these seven models, LTNet has only 32,614 parameters, which is less than the number of parameters of the LTNet (Original version). Specifically, it is equivalent to just 0.139% of the traditional convolutional neural network ResNet50, less than 0.1% of AlexNet and VGG16 and even less than 1% of the popular lightweight convolutional neural networks EfficientNet and MobileNetV3_large. The optimal training weight size of LTNet is 0.155 MB, demonstrating a more efficient memory utilization compared to MobileNetV3_large, EfficientNet, AlexNet and LTNet (Original version). This indicates that relative to the existing lightweight convolutional neural networks and traditional convolutional neural networks, LTNet is better suited for use in resource-constrained environments, and by comparing LTNet with its unimproved version, we have demonstrated that the improvements made to LTNet have a lightweight effect.

### 4.3. Classification Results of Own Mixed Gas Dataset

From Figure 5e–k and Table 5, it can be observed that for the own mixed gas dataset, LTNet has five errors in the mixed gas category and only two errors in the ethanol gas category, achieving the highest classification accuracy of 99.06%. Additionally, it is worth noting that LTNet’s GPU RAM usage during training is significantly lower than that of traditional convolutional networks, such as VGG16 and lightweight networks like EfficientNet. It completes thirty rounds of training in only 844.38 s, making it the fastest among all compared networks, far surpassing ResNet50, VGG16 and EfficientNet. LTNet only takes 23 s to complete inference on 744 validation set images, making it the fastest among these six networks. It significantly outperforms traditional convolutional neural networks, with the required inference time being only a quarter of that of the lightweight network MobileNetV3_large. Compared to LTNet (Original version), LTNet has even more advantages.

### 4.4. UCI Database Classification Results

From Figure 5l–r and Table 6, it can be observed that for the UCI database, LTNet achieves similar results as on the own mixed gas dataset. LTNet still maintains the highest classification accuracy while significantly outperforming the traditional convolutional neural network models and lightweight convolutional neural network models in terms of GPU RAM, training time and inference time.

Results from the own mixed gas dataset and the UCI database demonstrate that LTNet can achieve high accuracy in gas classification tasks while maintaining low computational resource requirements, further validating the lightweight nature of LTNet. Moreover, LTNet demonstrates higher accuracy than LTNet (Original version), with better lightweighting effects, validating the effectiveness of LTNet’s improvements.

## 5. Conclusions

In this study, we proposed a lightweight and efficient LTNet network model combined with GASF to convert one-dimensional time series into two-dimensional images for high-precision classification of acetone and ethanol gases, which are respiratory markers for lung cancer patients. The six evaluation metrics verified that LTNet outperforms classical convolutional neural network models, such as VGG16 and ResNet50, as well as lightweight neural network models, such as MobileNetV3_large. Validation with the own mixed gas dataset and the UCI database shows that compared with the other six models, LTNet has higher classification accuracy, superior generalization performance, and fewer parameters. By fusing the convolutional layer with the BN layer, the inference speed of LTNet in the validation set is much faster than that of ResNet50, MobileNetV3_large and so on. In addition, it required less graphic card resources during the training process and the model weights took up less memory. This indicated that the LTNet network requires less computational resources and is suitable for less configured hardware. The lightweight network model lays the foundation for subsequent algorithm transplant. At the same time, it verifies the effectiveness of the improvements made to LTNet. In the future, artificial intelligence and novel biomarkers could play a key role in the entire lung cancer screening process, promising to transform lung cancer screening [32].

## Figures and Tables

**Figure 1 sensors-24-02818-f001:**
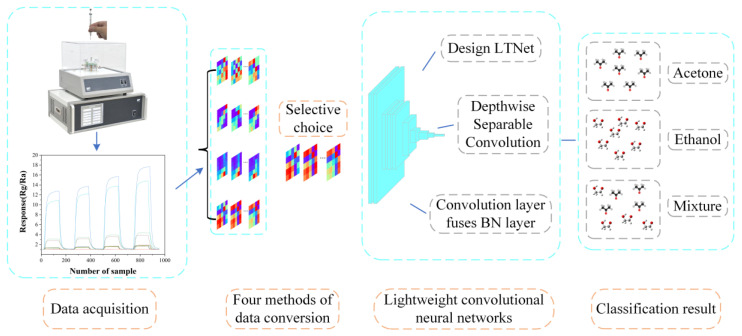
Schematic diagram of the overall process of this work.

**Figure 2 sensors-24-02818-f002:**
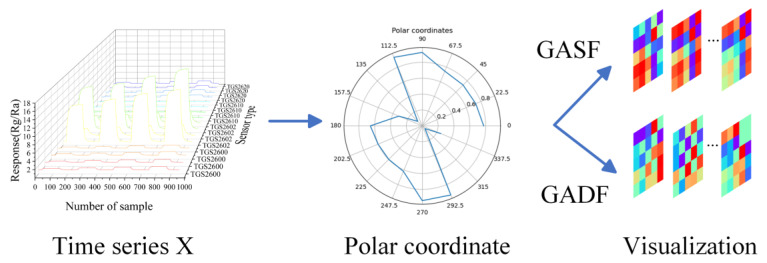
The diagram of GAF coding. X represents the response data of 16 sensors at a certain sampling point. After scaling, X is converted to the polar coordinate system, and, finally, the GASF/GADF image is generated.

**Figure 3 sensors-24-02818-f003:**
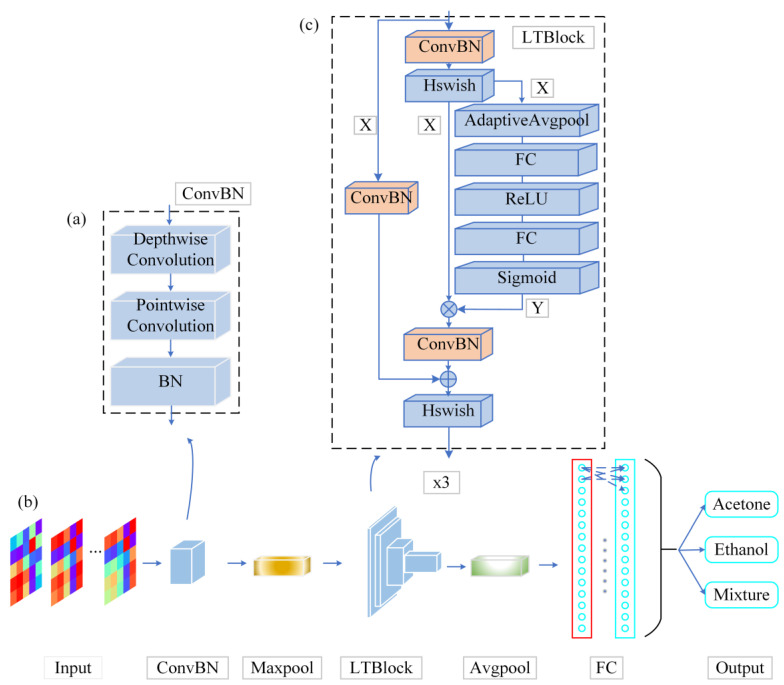
Block diagrams of LTNet network structure. (**a**) The ConvBN module is presented, which results from the fusion of depthwise separable convolution and batch normalization (BN) layers, (**b**) the backbone network and (**c**) the LTBlock module, constructed by integrating the ConvBN module, residual connections and the squeeze-and-excitation (SE) attention mechanism.

**Figure 4 sensors-24-02818-f004:**
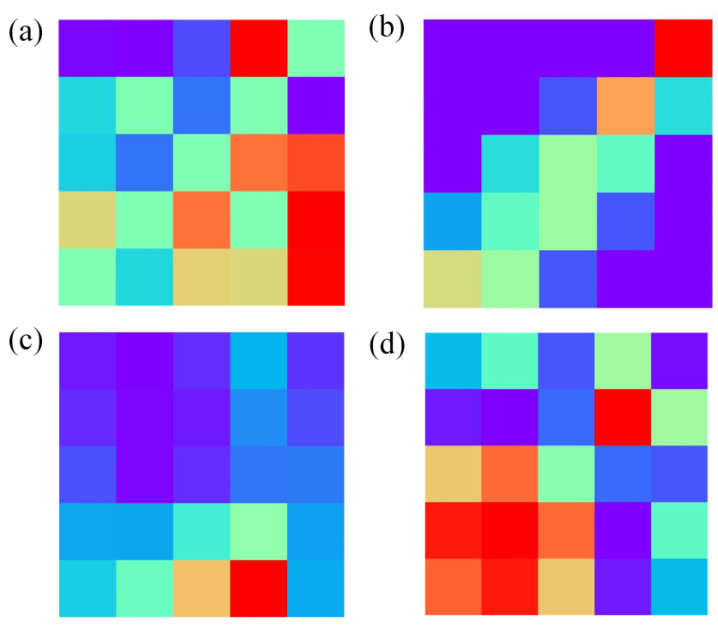
Image generated after data conversion. (**a**–**d**) Images converted by GADF, MTF, SFTF and GASF, respectively.

**Figure 5 sensors-24-02818-f005:**
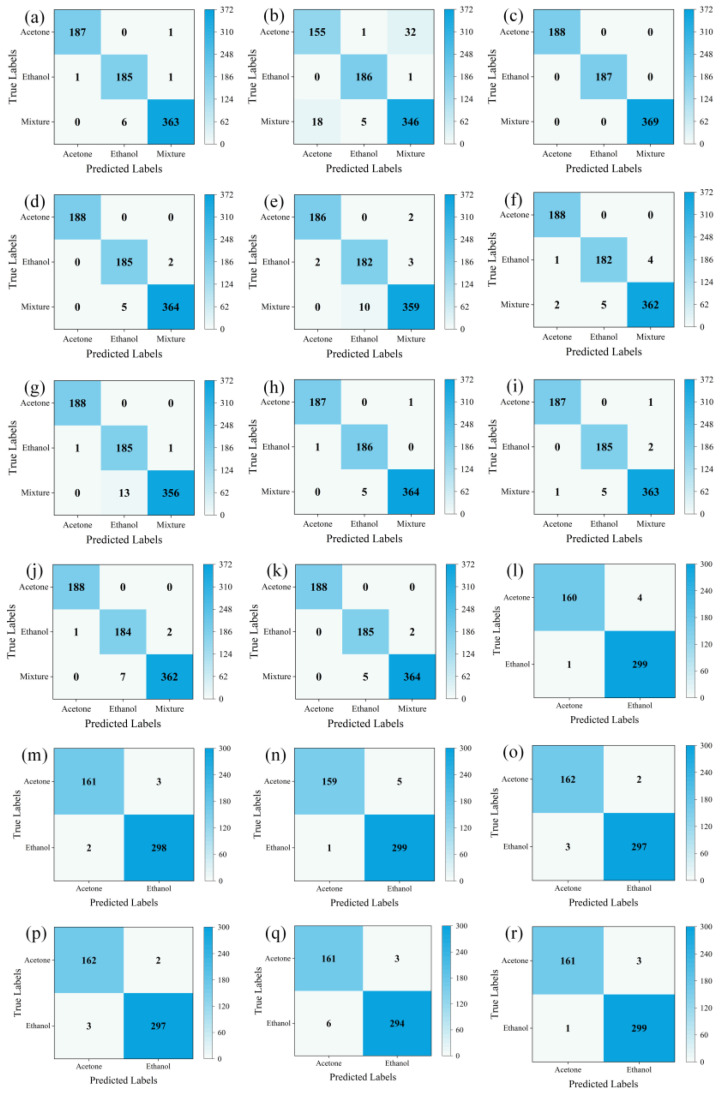
The confusion matrices. (**a**–**d**) The evaluation results of LTNet on the GADF, MTF, SFTF and GASF image transformation methods, respectively. (**e**–**k**) The evaluation results of AlexNet, ResNet50, VGG16, EfficientNet, MobileNetV3_large, LTNet (original version) and LTNet on the mixed gas dataset. (**l**–**r**) The evaluation results of AlexNet, ResNet50, VGG16, EfficientNet, MobileNetV3_large, LTNet (original version) and LTNet on the UCI database.

**Table 1 sensors-24-02818-t001:** Sensor models and optimal operating currents.

NO.	Models	Target Gases	Detection Ranges (ppm)	Optimal Operating Currents (mA)
1	TGS2600	Ethanol, Hydrogen	1–30	45
2	TGS2602	Ammonia, Ethanol	Ethanol 1–30	50
3	TGS2610	Organic compounds	500–10,000	55
4	TGS2620	Ethanol, Organic compounds	Ethanol 50–5000	43

**Table 2 sensors-24-02818-t002:** Details of concentration indicators in the acetone ethanol experimental dataset.

NO.	Ethanol (ppm)	Acetone (ppm)	Mixed Gas (ppm)
1	0	1	1
2	0	3	3
3	0	5	5
4	0	7	7
5	0	9	9
6	0	11	11
7	0	13	13
8	0	15	15
9	1	0	1
10	3	0	3
11	5	0	5
12	7	0	7
13	9	0	9
14	11	0	11
15	13	0	13
16	15	0	15
17	1	1	2
18	1	5	6
19	1	10	11
20	1	15	16
21	5	1	6
22	5	5	10
23	5	10	15
24	5	15	20
25	10	1	11
26	10	5	15
27	10	10	20
28	10	15	25
29	15	1	16
30	15	5	20
31	15	10	25
32	15	15	30

**Table 3 sensors-24-02818-t003:** Comparison of results of four data conversion methods.

Models	Accuracy	Training Time (S)	GPU RAM (G)
GADF	98.79%	863.49	2.3
MTF	92.34%	929.08	2.6
STFT	100%	1630.92	2.6
GASF	99.06%	844.38	2.1

**Table 4 sensors-24-02818-t004:** Model parameters and weights.

Models	Params.	Weight Size (MB)
AlexNet	57,012,034	217
ResNet50	23,514,179	89.9
VGG16	134,268,738	512
EfficientNet	4,586,092	17.8
MobileNetV3_large	4,208,443	16.2
LTNet (Original version)	296,994	1.15
LTNet (This work)	32,614	0.155

**Table 5 sensors-24-02818-t005:** Classification results of mixed gas datasets.

Models	Accuracy	GPU RAM (G)	Training Time (S)	Inference Time (S)
AlexNet	97.71%	3.1	853.27	283
ResNet50	98.39%	3.8	1234.34	284
VGG16	97.98%	6.9	2249.56	592
EfficientNet	99.06%	5.4	1373.48	170
MobileNetV3_large	98.79%	3.3	877.53	91
LTNet (Original version)	98.65%	2.3	1112.03	26
LTNet (This work)	99.06%	2.1	844.38	23

**Table 6 sensors-24-02818-t006:** Classification accuracy and model parameters of UCI database.

Models	Accuracy	GPU RAM (G)	Training Time (S)	Inference Time (S)
AlexNet	98.92%	3.2	613.50	187
ResNet50	98.92%	3.7	841.81	178
VGG16	98.71%	7.1	1477.44	377
EfficientNet	98.92%	5.3	933.21	109
MobileNetV3_large	98.92%	3.3	606.20	60
LTNet (Original version)	98.06%	2.3	859.49	18
LTNet (this work)	99.14%	2.1	584.67	14

## Data Availability

Data available on request from the authors.

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
