# Peer review of "The Classification of VOCs Based on Sensor Images Using a Lightweight Neural Network for Lung Cancer Diagnosis"

_sensors, 2024, doi:10.3390/s24092818_

Round 1

Reviewer 1 Report

Comments and Suggestions for Authors

This paper presents a lightweight CNN model that can discriminate VOCs with high accuracies. The work has its merits and would be of interest to researchers in the relevant fields. Therefore, I would recommend it for publications with few minor corrections.

(1) The sensor model used for testing, has it been prototyped? Or a commercial setup? Could the authors include setup figures of the unit?

(2) How is the experiment been conducted? Is it comparable to the open dataset from UCI?

(3) The gas concentrations for mixed gas are much higher than the single gases, thus the sensor unit will has a greater response towards the mixed gas, which would affect the accuracy of the CNN model, and would also explain why the accuracy for the mixed gas is higher than that of the single gases. Any concerns for that? Or perhaps results for mixed gas of the same concentration levels?

Reviewer 2 Report

Comments and Suggestions for Authors

As claimed by the authors, a lightweight and efficient LTNet network model was proposed for the high-precision classification of respiratory biomarkers (acetone, ethanol, and their mixture) in lung cancer patients. Furthermore, comparisons were made among GASF, GADF, STFT, and MTF, demonstrating that GASF achieves higher accuracy with shorter training times. Through comparisons with other classical networks using multiple parameters, it was demonstrated that LTNet network exhibits higher classification accuracy, better generalization performance, and fewer model parameters. In summary, this article is of great practical significance. However, it seems that more data are needed for comparison in the classification section. I have some comments or questions as followed, before considering this work for publishing.

1. The title mentions " using a lightweight neural network for lung cancer diagnosis ", but the text only involves using the network for classifying acetone, ethanol, and their mixture. It does not provide a detailed description of how respiratory biomarkers are utilized for lung cancer diagnosis. Supplementing this explanation would enhance the practical significance of the article.

2. In the introduction section where the innovative aspects of the article are presented, the proposed improvements to the neural network model have not been supported by data. Adding a comparison of data before and after the improvement of the network model in the data analysis section (similar to the comparison between LTNet and VGG16 networks) would enhance the persuasiveness of the article's innovation.

3. How much time is required from collecting breath samples from patients to completing lung cancer screening? What is the degree of association between respiratory biomarkers and lung cancer? In other words, what is the ultimate accuracy achievable in diagnosing lung cancer using patients' respiratory biomarkers?

Comments on the Quality of English Language

 Moderate editing of English language required
